# *Streptococcus thermophilus* alters the expression of genes associated with innate and adaptive immunity in human peripheral blood mononuclear cells

Narges Dargahi[1]*, Joshua Johnson[2], Vasso Apostolopoulos[1]*

**1** Institute for Health and Sport, Victoria University, Melbourne, Victoria, Australia, **2** Institute for Sustainable Industries and Liveable Cities, Victoria University, Melbourne, Victoria, Australia

* Narges.Dargahi@live.vu.edu.au (ND); Vasso.Apostolopoulos@vu.edu.au (VA)

## Abstract

Consumption of probiotics contributes to a healthy microbiome of the GIT leading to many health benefits. They also contribute to the modulation of the immune system and are becoming popular for the treatment of a number of immune and inflammatory diseases. The main objective of this study was to evaluate anti-inflammatory and modulatory properties of *Streptococcus thermophilus*. We used peripheral blood mononuclear cells from healthy donors and assessed modifications in the mRNA expression of their genes related to innate and adaptive immune system. Our results showed strong immune modulatory effects of *S. thermophilus* 285 to human peripheral blood mononuclear cells with an array of anti-inflammatory properties. *S. thermophilus* 285 reduced mRNA expression in a number of inflammatory immune mediators and markers, and upregulated a few of immune markers. *S. thermophilus* is used in the dairy industry, survives during cold storage, tolerates well upon ingesting, and their consumption may have beneficial effects with potential implications in inflammatory and autoimmune disorders.

## 1. Introduction

The human body and, in particular, the gastrointestinal tract (GIT) hosts a variety of microbial populations referred to collectively as the microbiome [1]. The microbiome of the GIT plays a key role in the maintenance of a healthy immune system [1, 2], and disruptions to the microbiome composition can lead to serious effects on health [3–5]. In order to maintain a healthy microbiome, regular ingestion of probiotic supplements, or the ingestion of fermented dairy products/capsules has been suggested. These practices have led to various improved health outcomes, ranging from enhanced overall human wellbeing to the treatment of infections, constipation, diarrhoea etc [1].

The majority of probiotics belong to the lactic acid bacteria (LAB) family; gram positive lactic acid producing microorganisms that include several genera such as bifidobacteria, lactobacilli streptococci and enterococci [1]. The small intestine and the colon are highly

**Funding:** The authors received no specific funding for this work.

**Competing interests:** The authors declare no competing interests.

enriched with these microorganisms [6–8], which are routinely supplemented in foods as live strains due to their beneficial effects on human health [1, 2, 8–13]. *Streptococcus* species such as exopolysaccharide-producing strains of *Streptococcus thermophilus* (ST) [12, 14, 15] are among those consumed. These characteristics of *S. thermophiles* enable them to be used in fermented milk products (i.e. yogurt) including flavoring of dairy, and is recognized as the next most important species after *Lactococcus lactis* [16, 17]. ST and *L. brevis* synergistically display well established health benefits, and *S. thermophilus* is one of the bacteria in the VSL#3 probiotic mixture, which has long been broadly applied in the treatment of inflammatory conditions [18, 19]. In addition, probiotics interact with the immune system leading to immunomodulation and anti-inflammatory properties [4, 20, 21].

The 'hygiene hypothesis' suggests that the positive trend in the incidence of immune-related disorders can been attributed to intestinal dysbiosis, resulting in immune dysfunction (ie. asthma, eczema, allergies and autoimmune diseases). Use of probiotic bacteria can increase abundance and concurrently modulate immune cells, including B, T helper (Th)-1, Th-2, Th-17 and regulatory T (Treg) cells. This in turn, directly influences human health and modulates pathologies of immune/autoimmune diseases [1, 2, 13]. In fact, we previously noted that *S. thermophilus* 1342, *S. thermophilus* 1275 and *S. thermophilus* 285 modulate the U937 monocyte cell line. Specifically, we showed that interleukin (IL)-4, IL-10, GM-CSF and CXCL8 production were increased, and, cell surface marker expression CD11c, CD86, C206, CD209, MHC-1 were upregulated [1]. In another study, *S. thermophilus* 1275 and *Bifidobacterium longum* BL536 demonstrated increased levels of transforming growth factor (TGF)-beta (a key factor in the differentiation of Treg and T-helper Th)-17 cells by bulk peripheral blood mononuclear cell (PBMC) cultures [22]. Primary macrophages co-cultured with ST bacteria stimulate production of anti-inflammatory IL-10 and pro-inflammatory IL-12 cytokines [23].

Peripheral blood mononuclear cells (PBMC) isolated from whole blood constitute a wide range of diverse immune cells that play vital roles in balancing immune homeostasis and keeping human health in check [24, 25]. These cells are crucial components of the innate and adaptive immune system, defend the body against bacterial, viral and parasitic infections, as well as destroying foreign antigens and cancer cells [25]. PBMC are predominantly made up of lymphocytes (~70–90%), monocytes (~10–20%) and other cells such as dendritic cells comprise less than 1–2% [26]. In spite of variations in the fraction of subtypes of immune cells within the total PBMC isolated from different samples [26], isolation, characterization and molecular studies of these cells have benefited medical research [27].

Herein, we describe changes in the expression of genes associated with innate and adaptive immunity including cytokines, chemokines and immune cell marker expression by human PBMC following exposure to live *S. thermophilus* 285 bacteria.

## 2. Material and methods

### 2.1. Bacterial strains

Pure bacterial cultures of *S. thermophilus* 285 were obtained from Victoria University culture collection (Werribee, VIC, Australia). Stock cultures were stored in cryobeads at −80˚C. Prior to each experiment the cultures were propagated in M17 broth (Oxoid, Denmark) with 20 g/L lactose and incubated at 37˚C under aerobic conditions. Bacteria were also cultured in M17 agar (1.5% w/v agar) with 20 g/L lactose (Oxoid, Denmark), to assess characteristics, morphology, purity and gram-positive confirmation [1].

## 2.2. Preparation of live bacterial suspensions

Media were prepared and autoclaved at 121˚C for 15 minutes (mins) prior to experiments. Bacterial cultures were grown 3 times in M17 broth with 20 g/L lactose, at 37˚C aerobically for 18 hours (hr) with a 1% inoculum transfer rate [28]. Cultures grow optimally at 37–42˚C for 24 hrs [15]. The growth period of cultures were consistent at 18 hr (at the end of the exponential growth phase) and before stationary growth phase to prevent cell lysis. Bacteria were harvested during stationary growth phase on the day of experiment, centrifuged (6000×g) for 15 min at 4˚C, followed by two washes with Dulbecco's phosphate-buffered saline (DPBS) (Invitrogen, Pty Ltd. Australia) and resuspended in the Roswell Park Memorial Institute (RPMI) 1640 culture media. These samples constituted the live-cell suspensions.

## 2.3. Enumeration of bacterial cells

Bacterial strains were scraped from M17 agar and transferred into Dulbecco's PBS (Invitrogen, Pty Ltd. Australia) adjusted to a final concentration of $10^8$ colony forming units (cfu)/ml by measuring the optical density at 600 nm, and washed two times with PBS and resuspended in RPMI 1640 prior to co-culturing with PBMC [1].

## 2.4. Isolation, culture, and stimulation of PBMC

**2.4.1. Isolation of PBMC using Ficoll-Paque.** PBMC isolation from whole blood was via Ficoll-Paque density gradient centrifugation [9]. Three buffy coats were collected from the Australian Red Cross Blood Bank on the day of experiment (Victoria University human research ethics). Calcium and magnesium free PBS, pH7.2, (Invitrogen, Pty Ltd. Australia) was used after adding 2 mM EDTA and 2% heat-inactivated fetal bovine serum (FBS) (Invitrogen, Pty Ltd. Australia); PBS buffer. SEPMATE tubes (50 ml) with inner inserts (STEMCELL technology, Canada) were used to isolate PBMC following Ficoll-Paque density gradient protocol [29, 30]. PBMCs were washed, counted and the required number of PBMC were co-cultured with *S. thermophilus* 285 and the remaining PBMC were stored in freeze mix and transferred into liquid nitrogen for future use.

**2.4.2. Stimulation of PBMC with *S. thermophilus* 285.** PBMC (3x $10^7$ cells) were resuspended in RPMI 1640 media supplemented with 10% heat-inactivated FBS (Invitrogen, Pty Ltd. Australia), 1% antibiotic-antimycotic solution and 2 mM L-glutamine in cell culture flasks, and 3x$10^8$ *S. thermophilus* 285 bacteria were added. PBMC with RPMI media without the addition of ST285 bacteria were used as a control and incubated at 37˚C, 5% $CO_2$ for 24 hrs [1]. We previously demonstrated that 24 hrs co-culture was optimal for stimulation of U937 monocyte/macrophage cell line, and all incubations described herein were for 24 hrs [1]. PBMCs were snap frozen post incubation and stored at -80˚C prior to RNA extraction.

## 2.5. RNA extraction from PBMC

Total RNA was extracted from stimulated PBMCs using the RNeasy® mini kit (Qiagen, Hilden, Germany) according to the manufacturer's instructions. Briefly, cells were centrifuged and harvested, supernatants were removed and RNA extracted from each cell pellet and resuspended in lysis buffer supplemented with β-mercaptoethanol to disrupt the cells. PBMC were lysed and each cell lysate passed through the supplied Qia-shredder columns to homogenize and was subsequently mixed with equal volume of 70% ethanol. Cell lysates were transferred onto RNeasy mini-spin columns and DNA was removed using DNase digestion/ treatment using RNase-Free DNase Set (Qiagen, Hilden, Germany.) The RNA Integrity Number (RIN) of all RNA samples were measured using an Agilent 2100 Bioanalyzer and Agilent RNA 6000

nano kit (Agilent Technologies, Santa Clara, CA, USA); with a minimum RIN of 7.5 used as the criterion for inclusion in gene expression analysis. The concentration of each individual RNA sample was measured using a Qubit RNA BR Assay (Invitrogen) in triplicate. Several blood samples were collected for PBMC isolation, treatment and extraction of RNA and only RNA samples with the highest RIN numbers (all above 8) were included for PCR.

## 2.6. Assessing changes in the expression of genes associated with innate and adaptive immunity

Aliquots of each RNA sample were reverse-transcribed to make complementary DNA (cDNA) using $RT^2$ first strand kit (Qiagen, Hilden, Germany) according to the manufacturer's instructions. Quantitative real-time polymerase chain reaction (qRT-PCR) was performed using the 'Human Innate and Adaptive immune Response' kit (Qiagen, Hilden, Germany) to evaluate gene/mRNA expression. The relative expression profiles of treated PBMC samples were analyzed in comparison with untreated PBMC cultured in RPMI using Thermo-cycler (Biorad, Melbourne Australia). The $RT^2$ qPCR Primer innate and adaptive immune response arrays target a set of 84 innate and adaptive immune-related genes and five housekeeping genes, an RT control, a positive PCR control, and a human genomic DNA contamination control. The levels of the expression of these genes were calculated using the Qiagen web-based software (Qiagen, Germany) and then calculated the fold changes and analyzed data manually to compare results. Differential expression (up and down regulation) of the genes were identified using the criteria of a $> 2.0$-fold increase/decrease in gene expression in treated PBMCs in comparison with those genes in control PBMC cultures.

## 2.7. Data analysis

The Delta-Delta CT ($\Delta\Delta$CT) was used to calculate fold-changes [31]. Fold-regulation represents fold-change results in a biologically meaningful way. In our RT2 profiler PCR array results, fold-change values greater than one, indicate a positive (or an up-) regulation, in fact in upregulated genes, the fold-regulation is equal to the fold-change. Fold-change values less than one specifies a negative (or a down) regulation, and in this case, the fold-regulation is the negative inverse of the fold-change [32–34]. Data related to changes in the expression of the genes were analyzed by $\Delta\Delta$CT method using Qiagen $RT^2$ profiler data analysis webportal that utilises the delta delta CT method in determining fold-changes. The raw CT values were uploaded to the Qiagen data analysis webportal with the lower limit of detection set for 35 cycles and 3 internal controls: PCR array reproducibility, RT efficiency and genomic DNA contamination were assessed to ensure all arrays successfully passed all of these control checks. Normalization of the raw data was performed using the included housekeeping genes (HKG) panel. Then using the $\Delta\Delta$CT method, both housekeeping gene references and untreated/ controls were assessed to calculate relative expression of mRNA.

## 2.8. Statistical analysis

The p values are calculated based on a Student's *t-test* of the Triplicate $2^{\wedge}$ (- Delta CT) $[(2^{\wedge}-\Delta CT)]$ values for each gene in the treatment group vs. the control group [32, 33, 35, 36].

## 3. Results

Among 84 genes assessed, 31 genes were significantly altered $> 2.0$ fold up/down in PBMC samples (n = 3) following exposure to *S. thermophilus* 285 compared to control PBMC (Fig 1, Table in S1 Table and S1 Fig).

**A**

| Layout | 01 | 02 | 03 | 04 | 05 | 06 | 07 | 08 | 09 | 10 | 11 | 12 |
|---|---|---|---|---|---|---|---|---|---|---|---|---|
| A | APCS -1.05 C | C3 -3.38 A | CASP1 -1.24 | CCL2 -1.88 | CCL5 -1.10 | CCR4 -1.22 | CCR5 -6.29 A | CCR6 -1.29 B | CCR8 -1.11 B | CD14 -25.29 A | CD4 -1.86 B | CD40 -15.39 |
| B | CD40LG -1.55 A | CD80 -1.70 A | CD86 -8.04 A | CD8A -2.96 C | CRP -1.05 C | CSF2 130.35 A | CXCL10 -5.30 A | CXCR3 -1.22 B | DDX58 -1.02 | FASLG 1.02 | FOXP3 -1.85 | GATA3 -22.15 A |
| C | HLA-A -1.45 | HLA-E -1.22 | ICAM1 -1.31 | IFNA1 -1.05 C | IFNAR1 -1.88 | IFNB1 -1.05 C | IFNG 8.72 A | IFNGR1 -4.03 | IL10 2.05 | IL13 1.52 B | IL17A -1.05 C | IL18 -73.04 A |
| D | IL1A 2.78 | IL1B 4.82 | IL1R1 -1.50 | IL2 -7.27 B | IL23A 3.08 A | IL4 -1.34 B | IL5 -1.45 B | IL6 25.12 | CXCL8 11.26 | IRAK1 -1.05 B | IRF3 1.08 | IRF7 -12.32 |
| E | ITGAM -2.76 A | JAK2 -1.29 | LY96 -1.85 | LYZ -37.91 | MAPK1 -1.79 | MAPK8 -1.27 | MBL2 -1.05 C | MPO -2.33 | MX1 1.17 | MYD88 -1.73 | NFKB1 -1.24 | NFKBIA -1.48 |
| F | NLRP3 -2.11 A | NOD1 -1.23 A | NOD2 -1.41 B | RAG1 1.05 B | RORC -1.70 B | SLC11A1 -4.72 | STAT1 1.47 | STAT3 -1.39 | STAT4 -1.26 | STAT6 1.02 | TBX21 -1.01 | TICAM1 -1.26 |
| G | TLR1 -2.63 A | TLR2 -2.68 | TLR3 1.36 B | TLR4 -5.65 A | TLR5 -1.30 B | TLR6 -1.73 | TLR7 1.07 B | TLR8 -11.41 A | TLR9 -1.29 B | TNF 6.10 | TRAF6 -1.18 | TYK2 -10.03 A |

**B**

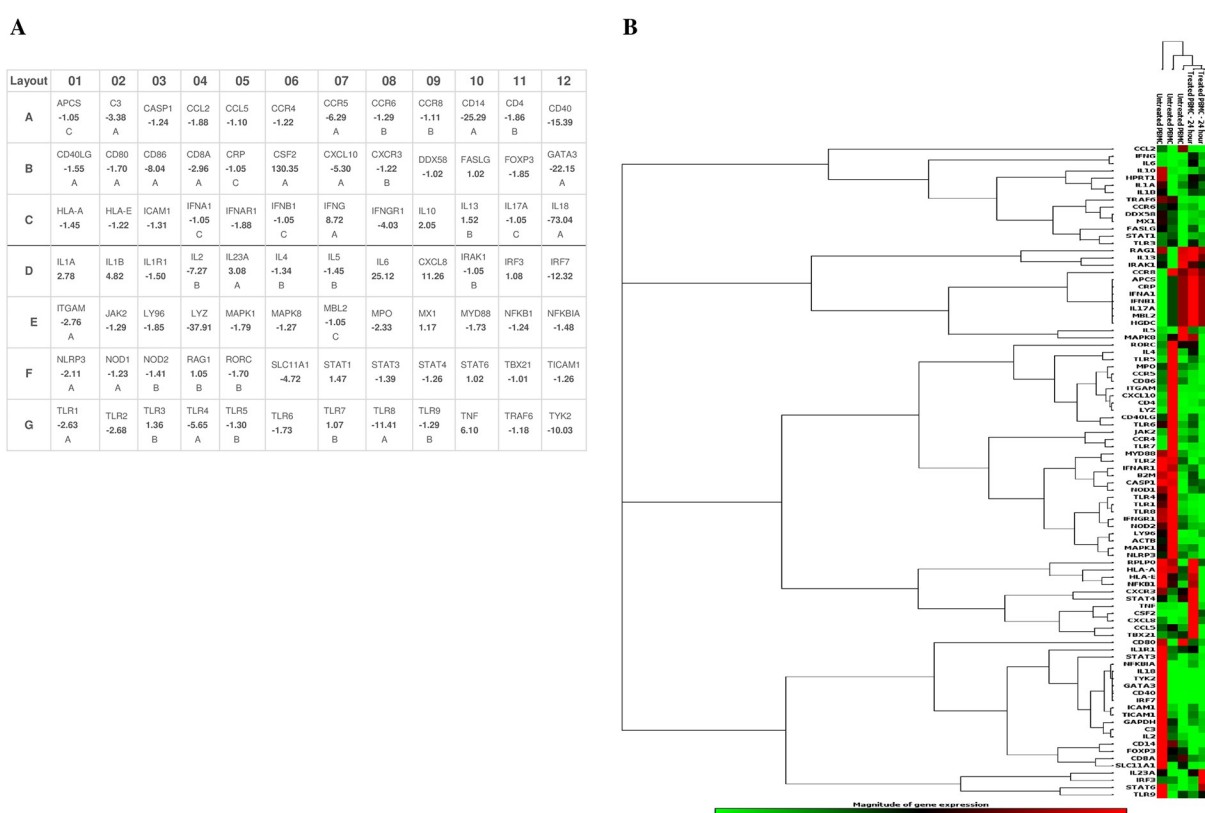

**Fig 1. Effects of co-culturing *S. thermophilus* 285 with PBMCs (n = 3) on gene/RNA expression compared to control PBMCs after 24 hrs.**
(A) All 84 genes are shown including those with significant high up/down regulated genes (more than 2-fold) and those with no significant change (less than 2-fold). The housekeeping genes (HKG) panel and other genes used for normalization of the raw data are not presented. Letter A specifies the gene's average threshold cycle to be reasonably high ($> 30$) in either the treated samples or the controls and relatively low ($< 30$) in the other/opposite sample. Thus, in case of presenting fold changes with letter A, the estimate fold change may be an underestimate. Letter B suggests a reasonably high ($> 30$) gene's average threshold cycle that means a low level of average expression of relevant gene, in both test/treated samples and untreated control samples, and the p-value for the fold-change might be either relatively high ($p > 0.05$). Thus, in case of presenting fold changes with letter B, the estimate fold change may be slightly overestimate or unavailable. Letter C indicates that that gene's average threshold cycle is either not determined or greater than the defined default 35 cut-off value, in both test/treated samples and control samples, suggesting that its expression was not detectable, resulting in the fold-change values being un-interpretable [86, 87] [88]. (B) Presentation of data as a hierarchical clusters of average gene/RNA expressions of PBMC (n = 3) co-cultured with *S. thermophilus* 285, compared to control. Green represents down regulated genes to red represents upregulated genes.

### 3.1. *S. thermophilus* 285 alters cytokine gene expression levels of PBMC

**3.1.1. Interleukin mRNA expression levels.** IL-1α and IL-6 are secreted by dendritic cells (DC), B cells and macrophages (MQ) are involved in acute phase responses, B cell maturation, macrophage differentiation, promote Th2 differentiation and inhibit Th1 polarization. IL-1α is upregulated 2.78 ± 0.6 fold and IL-6 25.12 ± 0.61 fold (Fig 2). IL- 23α is secreted by CD4+ T cells and aids in the stimulation of Th17 cells together with IL-6. IL-23α is highly upregulated 3.8 ± 1.0 fold (Fig 2). IL-2 has an array of functions it activates T cell proliferation and increases or decreases inflammatory responses. IL-2 is downregulated 7.27 ± 0.53 fold (Fig 2). IL-17A a pro-inflammatory cytokine secreted by Th17 cells, was not altered following PBMC co-cultured with *S. thermophilus* 285.

**3.1.2. Th1/Th2 mRNA expression levels.** IFNγ, a Th1 cytokine important in the defense against bacterial infection is upregulated 8.73 ± 0.94 fold. Likewise, the Th1 cytokine IL-1β is upregulated 4.82 ± 0.74 fold (Fig 3). Of interest, IL-18 a Th1 inducing pro-inflammatory

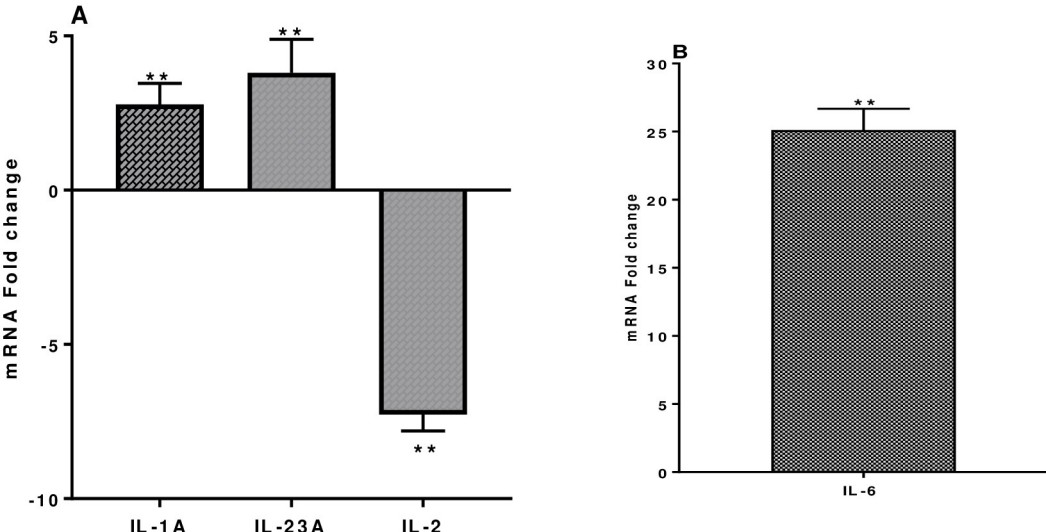

**Fig 2. (A) IL-1α, IL-23α and IL-2 and (B) IL-6, mRNA fold change following 24 h co-culture of *S. thermophilus* 285 with PBMCs (n = 3), compared to control PBMC.** The innate and adaptive RT$^2$ gene profiler arrays were used to determine changes in gene expression. Symbols represent *p* value for Tukey Test (One way ANOVA) where $^{**}$ $p < 0.04$.

cytokine was vastly downregulated (75 ± 0.66 fold), in addition, IFNγR1, a transmembrane protein which interacts with IFNγ, is also downregulated 4.03 ± 0.25 fold (Fig 3). Tumor-necrosis factor-alpha (TNFα), important in the defense against bacterial infections, and in acute phase reactions is upregulated 6.10 ± 1.4 fold (Fig 3). IL-10, an anti-inflammatory

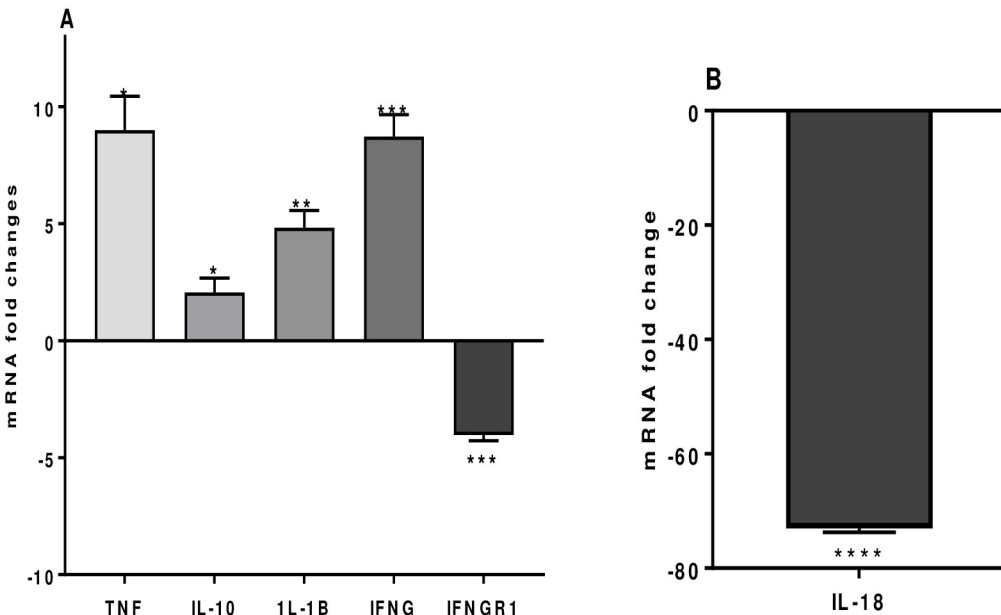

**Fig 3. A) TNF-α, IL-10, IL-1β, IFN-γ, and IFN-γ–R and (B) IL-18, mRNA fold change following 24 h co-culture of *S. thermophilus* 285 with PBMCs (n = 3), compared to control PBMC.** (The innate and adaptive RT$^2$ gene profiler arrays were used to determine changes in gene expression. Symbols represent *p* value for Tukey Test (One way ANOVA) where $^{*}$ $p < 0.05$, $^{**}$ $p < 0.04$, $^{***}$ $p < 0.02$ and $^{****}$ $p < 0.01$.

cytokine secreted by Th2 and Treg cells is upregulated $2.05 \pm 0.52$ fold (Fig 3). Gene expressions of other cytokines, IFNB1, IL-4, IL-5 and IL-13 are not significantly altered.

### 3.2. *S. thermophilus* 285 alters chemokine gene expression levels of PBMC

Chemokine (CXCL8, IL-8) is important in the innate immune system, it stimulates chemotaxis and is upregulated $11.26 \pm 0.27$ fold following *S. thermophilus* 285 co-culture with PBMC cells. However, CCR5 and CXCL10 (INP10) are down regulated $6.29 \pm 0.32$ and $5.30 \pm 1.8$ fold respectively (Fig 4). No significant differences are noted for gene expressions of other chemokines, including CCL2 (MCP-1), CCL5 (RANTES), CCL8, CCR4, CCR8, CXCR3, CCL2, IFNA1.

### 3.3. Colony stimulating factor mRNA expression levels

Colony-stimulating factor (CSF)-2, secreted by MQs, NK cells and T cells, enables cell proliferation and differentiation and is significantly increased by $130.35 \pm 1.0$ fold (Fig 5) after co-culturing PBMC with *S. thermophilus* 285 bacteria.

### 3.4. *S. thermophilus* 285 alters Toll like receptor gene expression levels of PBMC

TLR (toll like receptor)-1, TLR-2, TLR-4 and TLR-8 are part of the innate immune response and involved in the defense response to bacteria. PBMC co-cultured with *S. thermophilus*

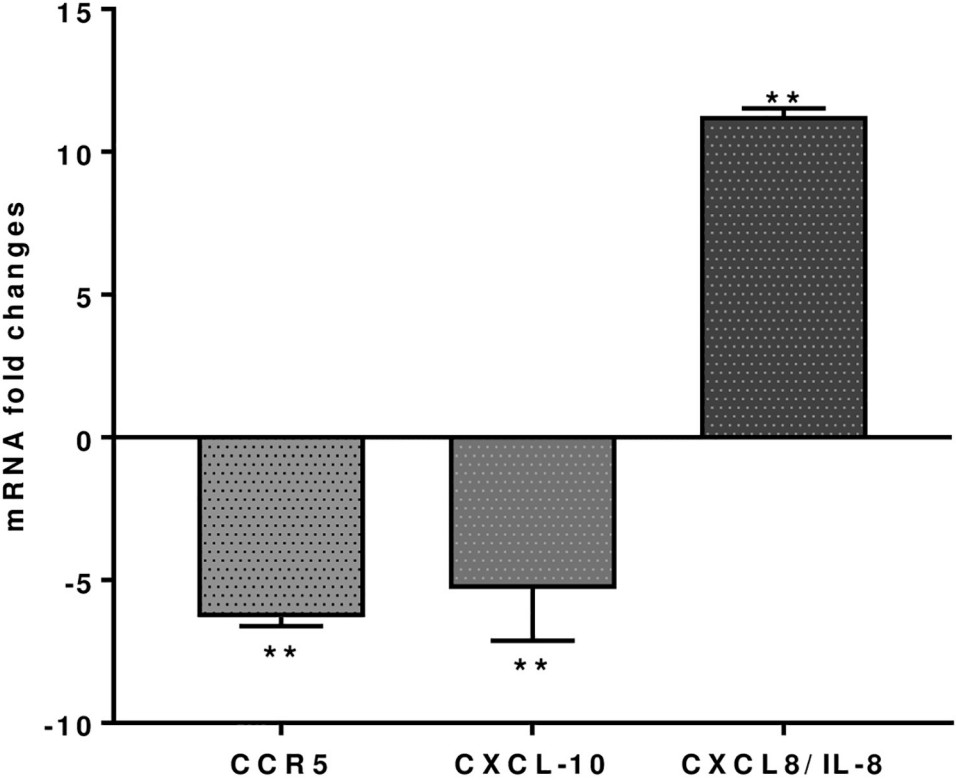

**Fig 4. CCR5, CXCL10 and CXCL8 (IL-8), mRNA fold change following 24 h co-culture of *S. thermophilus* 285 with PBMCs (n = 3), compared to control PBMC.** The innate and adaptive RT$^2$ gene profiler arrays were used to determine changes in gene expression. Symbols represent *p* value for Tukey Test (One way ANOVA) where $^{**}$ $p < 0.04$.

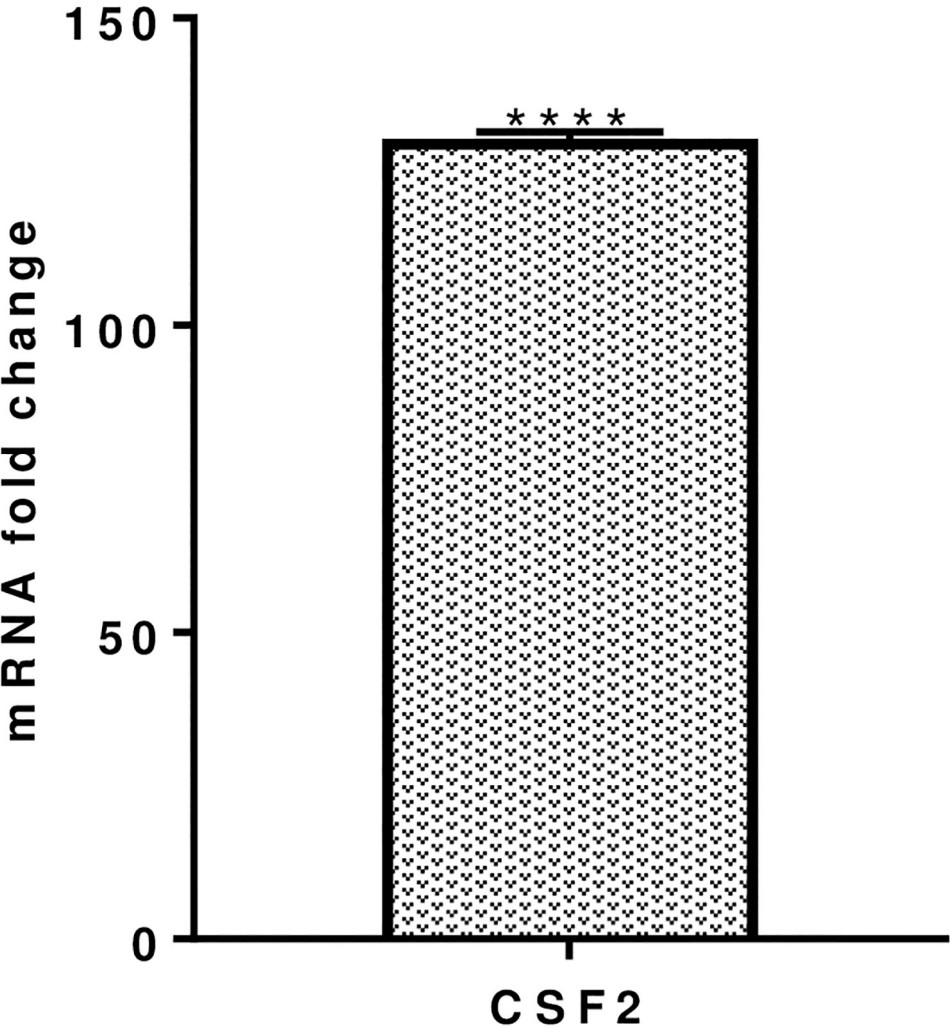

**Fig 5. CSF-2, mRNA fold change following 24 h co-culture of *S. thermophilus* 285 with PBMCs (n = 3), compared to control PBMC.** The innate and adaptive RT² gene profiler arrays were used to determine changes in gene expression. Symbols represent *p* value for Tukey Test (One way ANOVA) where **** p < 0.01.

285 induced downregulation of TLRs at varying levels; TLR-1 (-2.63 ± 0.43), TLR-2 (-2.69 ± 0.8 fold), TLR-4 (-5.65 ± 0.56 fold), TLR-8 (-11.41 ± 1.27 fold) (Fig 6). However, changes to other pattern recognition receptors such as, TLR-3, TLR-5, TLR-6, TLR-9 were not significant.

### 3.5. Cell surface markers CD14, CD40, CD86 mRNA expression levels

Expression of the monocyte cell surface markers CD14, CD40 and CD86 significantly downregulated -25.29 ± 3.46, -15.39 ± 1.36, -8.04 ± 0.14 fold, respectively (Fig 7). Expression of the CD8A gene, which is involved in adaptive immunity and in response to defense against viruses, was downregulated by -2.96 ± 0.68 fold (Fig 7). Expression of CD4, CD80, FOXP3, STAT3, CD40LG (TNFSF5), HLA-A, HLA-E and RORC genes do not show significant changes.

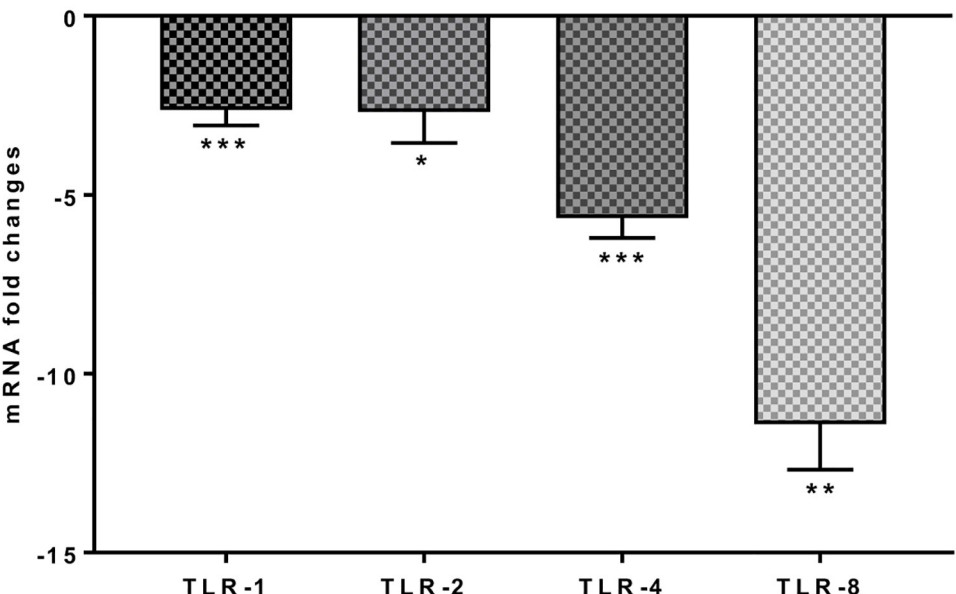

**Fig 6. TLR-1, TLR-2, TLR-4 and TLR-8, mRNA fold change following 24 h co-culture of *S. thermophilus* 285 with PBMCs (n = 3), compared to control PBMC.** The innate and adaptive RT[2] gene profiler arrays were used to determine changes in gene expression. Symbols represent *p* value for Tukey Test (One way ANOVA) where * $p < 0.05$, ** $p < 0.04$ and *** $p < 0.02$.

## 3.6. Changes to other innate and adaptive molecules, mRNA expression levels

Changes to other genes were also noted following *S. thermophilus* 285 co-culture with PBMC. ACTB (-3.01 ± 1.0) fold, ITGAM (-2.76 ± 0.9) were both downregulated. Downregulated genes were noted to the following: MPO (2.33 ± 0.2), NLRP3 (2.11 ± 0.6), SLC11A1 (4.72 ± 0.23) and complement component (C)-3 (3.38 ± 1.5), TYK2 (10.03 ± 0.7), IRF7

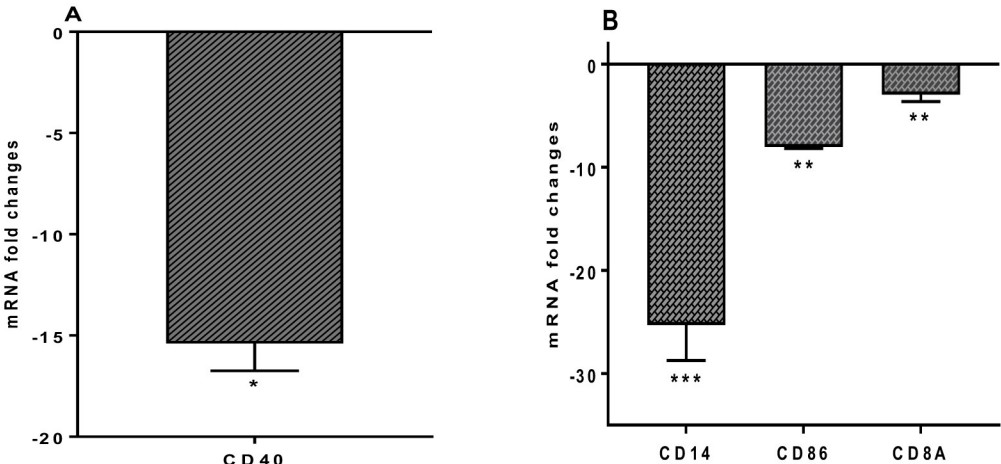

**Fig 7. (A) CD40 and (B) CD14, CD86 and CD8A, mRNA fold change following 24 h co-culture of *S. thermophilus* 285 with PBMCs (n = 3), compared to control PBMC.** The innate and adaptive RT[2] gene profiler arrays were used to determine changes in gene expression. Symbols represent *p* value for Tukey Test (One way ANOVA) where * $p < 0.05$, ** $p < 0.04$ and *** $p < 0.02$.

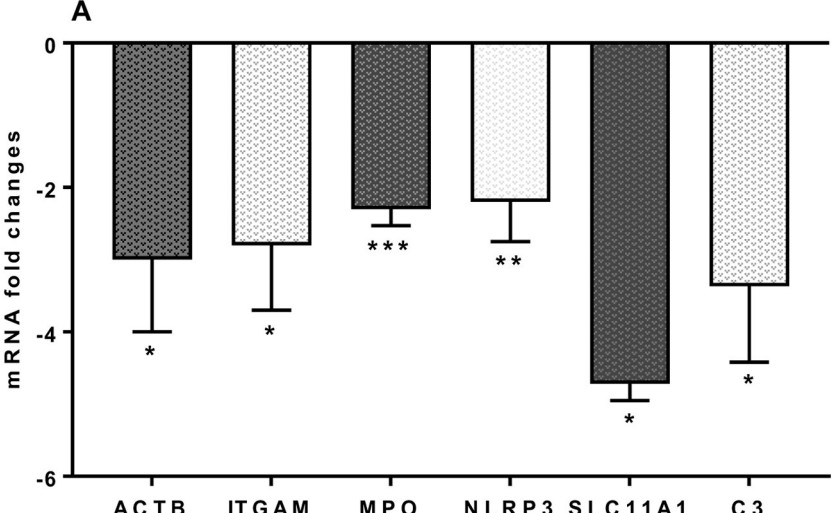

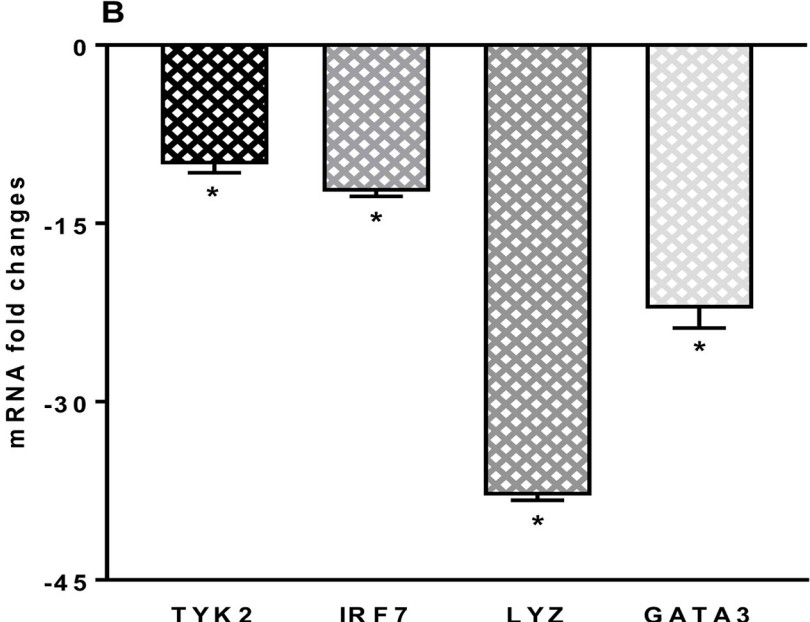

**Fig 8. (A) ACTB, CCR5, ITGAM, MPO, NLRP3, SLC11A1, and C3 and (B) TYK2, IRF7, LYZ and GATA3, mRNA fold change following 24 h co-culture of *S. thermophilus* 285 with PBMCs (n = 3), compared to control PBMC.** The innate and adaptive RT$^2$ gene profiler arrays were used to determine changes in gene expression. Symbols represent *p* value for Tukey Test (One way ANOVA) where * $p < 0.05$, ** $p < 0.04$ and *** $p < 0.02$.

(12.32 ± 0.4), LYZ (37.91 ± 0.4) and GATA3 (22.15 ± 1.64) (Fig 8). Other immune markers including FASLG (TNFSF6), CRP, IFNAR1, JAK2, IL-1R1, MAPK8 (JNK1), IRF3, MBL2, NFKB1, MX1, ICAM1, MBL2, MYD88, NOD1 (CARD4), NOD2, DDX58 (RIG-I), RAG1 and TICAM1 (TRIF) showed no significant mRNA gene changes in the levels of their expression.

## 4. Discussion

Our previous publications illustrated immune modulatory effects of *S. thermophilus* 285, *S. thermophilus* 1275 and *S. thermophilus* 1342 on U937 monocytic cell line and human monocytes by using secreted cytokines for bioplex assays, as well as flow cytometry of immune cell surface marker changes. The current study, aimed to get a more comprehensive overview of the data, by undertaking an in depth gene array analysis of the effects of probiotics to human PBMC.

### 4.1. *S. thermophilus* 285 promotes Th2 polarization

IL-1α secreted by peripheral blood DC and B cells induces Th2 differentiation and inhibits Th1 polarization [37], is significantly upregulated. Similarly, *Enterococcus faecium* NCIMB 10415 was shown to upregulate IL-1α in porcine jejunal epithelial cells (IPEC-J2) *in vitro*, [38]. IL-6 produced by Th2 cells is increased in the presence of *S. thermophilus* 285 by PBMC which was also shown previously to be upregulated by pro-monocyte cell line U937 [1]. Others have shown that PBMC co-cultured with *S. thermophilus* 1275 also increases IL-6 [39]. Likewise, mixed probiotics of *S. thermophilus*, *Lactobacillus* (*L.*) *rhamnosus*, *L. casei*, *L. acidophilus*, *B. longum* and *B. bifidum* stimulated PBMC to produce IL-6 [40, 41]. Our study shows that IL-1α and IL-6 are increased, highlighting the role of *S. thermophilus* 285 in stimulation of immune responses involved in acute phase; B cell maturation, macrophage differentiation, promotion of Th2 differentiation and inhibiting Th1 polarization.

IL-10 is an anti-inflammatory cytokine secreted by Th2 and Treg cells and co-culture of *S. thermophilus* 285 with PBMC increased expression of IL-10. Cultured PBMC with other live *S. thermophilus* strain (*S. thermophilus* 1275) also showed increased IL-10 [22, 39, 42–45]. Similarly, in a study using mixed probiotic cultures (*S. thermophilus*, *L. rhamnosus*, *L. casei*, *L. acidophilus*, *B. longum* and B. *bifidum*) high levels of IL-10 were stimulated by PBMC [40]. Conversely, in a study using *B. breve* and *S. thermophilus* combined to stimulate PBMC, IL-10 was only increased in the presence of *B. breve*, whereas exposing PBMC to *S. thermophilus* reduced the IL-10 level [46]. We also previously noted that monocyte cell line (U937), co-cultured with *S. thermophilus* 1342 stimulated production of high levels of IL-10 [1].

IL-18 is involved in the initiation of severe inflammatory responses, indicating the role of IL-18 in inflammatory and autoimmune disorders. Co-culture of PBMC with *S. thermophilus* 285 significantly downregulated IL-18 which indicates an anti-inflammatory role for *S. thermophilus* 285 bacteria. Likewise, a mixture of Lactobacilli species (*L. rhamnosus*, *L. paracasei*, and *L. plantarum*) was shown to supress the secretion of pro-inflammatory IL-18 gene by undifferentiated IPEC-1 intestinal porcine epithelial cell line [47], highlighting supportive role of lactobacilli probiotics in functioning against inflammation and suppression of immune response activities. However, other studies with other probiotics such as, *L. rhamnosus* E509, *L. rhamnosus* GG E522 (ATCC 53103), *L. bulgaricus* E585 and *S. pyogenes* serotype T1IH32030, increased IL-18 production by human PBMC [48]. Hence, different probiotic strains induce different cytokine profiles.

IL-2 is involved in signalling of immune responses and activates proliferation of lymphocytes. We note downregulation of IL-2 gene expression in PBMC after exposure to ST285. IL-23 known to activate Th17 cells was upregulated although IL-17, the key pro-inflammatory cytokine secreted by Th17 cells was not altered. Upregulation of IL-1α, IL-6, IL-10, and downregulation of IL-2, IL-18 and an absence of change in IL-17A (despite increase in IL-23α) designates ST285 to possess anti-inflammatory effects on human PBMC.

## 4.2 *S. thermophilus* 285 stimulates expression of cytokines involved in the defence against bacteria

IFN-γ is an adaptive immunity cytokine secreted by Th1 cells in the defense response to microbes and viruses. IFN-γ is predominantly secreted by NK, NKT cells as part of the innate immune response, and by CD4 Th1 and CD8+ T cells of the adaptive immune response [49]. *S. thermophilus* 285 upregulated IFN-γ gene expression by human PBMCs. This is similar to studies of a combination of probiotic strains including *S. thermophilus*, *Lactobacillus*, *Bifidobacterium*, *Propionibacterium*, *E. coli* and *Leuconostoc* [50], where upregulation of IFN-γ mRNA expression by PBMC was noted [50]. Likewise, co-cultures of pooled PBMC with ST1275 also induced upregulation of IFN-γ [39]. We previously noted that monocyte cell line co-cultured with *S. thermophilus* 1342, *S. thermophilus* 1275 or *S. thermophilus* 285 strains induced high levels of IFN-γ secretion [1]. In a study with Lactobacilli (*L. rhamnosus* E509, *L. rhamnosus* GG E522 (ATCC 53103) and *L. bulgaricus* E585), and streptococci (*S. pyogenesserotype* T1 IH32030), IFN-γ was produced by human PBMC [48].

IL-1β secretion by monocytes is involved in regulating immune and inflammatory responses to bacterial infections and injury, hence its role in innate immunity [51]. IL-1β is upregulated by *S. thermophilus* 285 co-cultured with PBMC, which is in accord with studies of PBMC co-cultured with mixed probiotics (*S. thermophilus*, *L. rhamnosus*, *L. casei*, *L. acidophilus*, *B. longum* and *B. bifidum*) [40]. We previously noted in monocyte cell line co-cultured with three different strains of *S. thermophilus*, only *S. thermophilus* 1342 stimulated production of high levels of IL-1β whereas, *S. thermophilus* 1275 and *S. thermophilus* 285 did not induce IL-1β cytokine [1]. A mixture of Lactobacilli strains (*L. rhamnosus*, *L. paracasei*, and *L. plantarum*) co-cultured with intestinal porcine epithelial cell line (IPEC-1) also upregulated IL-1β gene expression [47]. Similarly, the combination of *L. casei* Shirota, *L. rhamnosus* GG, *L. plantarum* NCIMB 8826 and *L. reuteri* NCIMB 11951, *B. bifidum* MF 20/5 and *B. longum* SP 07/3 co-cultured with PBMC, significantly augmented IL-1β production [41].

TNFα plays a key role in the defense against bacterial infections. It is a pro-inflammatory cytokine, which also supports recruitment and activation of T and B cells to promote an adaptive immune response. We previously demonstrated high levels of TNFα secretion by U937 monocyte cell line in the presence of *S. thermophilus* 1342, *S. thermophilus* 1275 and *S. thermophilus* 285 [1]. Likewise, our current findings show that ST285 co-cultured with PBMC results in upregulation of TNFα. However, in a study using *B. breve* and ST together to stimulate PBMC, TNF-α secretion was inhibited [46]. In addition, a mixture of strains of probiotics (*L. casei* Shirota, *L. rhamnosus* GG, *L. plantarum* NCIMB 8826 and *L. reuteri* NCIMB 11951, *B. bifidum* MF 20/5 and *B. longum* SP 07/3) co-cultured with PBMC, significantly increased the production of TNFα [41]. In another study of human PBMCs exposed to different probiotics (*L. mesenteroides* ssp. cremoris PIA2 (DSM 18892) *S. pyogenes* serotype T1M1, *S. thermophilus* THS, *E. coli* (DH5α), *L. rhamnosus* Lc705 (DSM 7061), *L. lactis* ssp. cremoris ARH74 (DSM 18891), *L. rhamnosus* GG (ATCC 53103), *L. helveticus* Lb 161, *L. helveticus* 1129, *B. longum* 1/10, *B. breve* Bb99 (DSM 13692), *B. animalis* ssp. lactis Bb12, and *Propionibacterium* (*P.*) *freudenreichii* ssp. shermanii JS (DSM 7067)), all induced TNF-α mRNA expression [50]. Given that IFNγ, IL-1β and TNFα are upregulated by PBMC following co-culture with *S. thermophilus* 285 this suggests that *S. thermophilus* 285 induces powerful defense against invading pathogens and could be beneficial against virus infection and tumours.

The upregulation of IFNγ, IL-1β and TNFα coupled with a significant decrease in IFNγ receptor and IL-18 shows an antagonising effect of *S. thermophilus* 285 inflammatory responses and leading to an overall anti-inflammatory profile.

## 4.3. *S. thermophilus* 285 activates mRNA expression of CXCL8 and downregulates CCR5 and CXCL10

IL-8, also known as CXCL8 is an important chemokine of the innate immune system, involved in the recruitment of neutrophils and other granulocytes as the first line of defense [52]. *S. thermophilus* 1342, *S. thermophilus* 1275 and *S. thermophilus* 285 were previously shown to activate U937 monocyte cell line to produce high levels of IL-8 [1]. The probiotic *L. paracasei* DG also increases expression of IL-8 to the human monocyte cell line, THP-1 [53]. Likewise, short chain fatty acids, produced by probiotic bacteria, also stimulate IL-8 secretion and mRNA levels to the human epithelial cell line HT-29 [11]. These studies are in accord to our current findings that *S. thermophilus* 285 upregulates CXCL8 production by human PBMC.

C-C chemokine receptor type 5 (CCR5, CD195) is involved in Th1 immune responses and its gene expression is downregulated by PBMC following *S. thermophilus* 285 co-culture. However, in mice prolonged feeding with VSL#3 probiotic mixture shows significant gene upregulation of CCR5 [54]. Differences could be attributed to one probiotic strain applied and varying effects of the strain (*S. thermophilus*) used in current study versus a mixture of different strains and species used in mice VSL#3 (*L. delbruekii Bulgaricus*, *L. casei*, *L. plantarum*, *L. acidophilus*, *B. breve*, *B. longum*, *B. infantis* and *S. thermophilus*).

CXC motif chemokine 10 (CXCL10), or IFN-γ-induced protein-10 (IP-10), is secreted by a number of cell types (endothelial cells, monocytes and fibroblasts). Few roles have been ascribed to CXCL10 including chemo-attraction of NK cells, monocytes/macrophages, T cells and DCs, favouring adhesion of T cells to endothelial cells, anti-cancer/tumour action, and preventing angiogenesis and bone marrow colony development. CXCL10 is downregulated in PBMC culture following *S. thermophilus* 285 exposure. Conversely, monocyte-derived DCs co-cultured with *B. breve* Bb99, *L. lactics* subsp. cremoris ARH74 and *S. thermophilus* THS increased expression of CXCL10 and *S. thermophilus* was the most efficient probiotic in the induction of CXCL10 [23]. Additionally, microarray results of the intestines of mice prolonged administrated with VSL#3 probiotic mixture in healthy mice showed differential effects on intestinal immune parameters, including upregulation of CXC10 which contrasts with our findings [54]. The difference are most likely due to cell types, as well as bacterial strains in our study (PBMC co-cultured with *S. thermophilus* 285 bacteria) compared to using mouse cells exposed to three strains (*B. breve* Bb99, *L. lactics* subsp. cremoris ARH74 and *S. thermophilus* THS) in the other study. Also in the latter experiments, it is quite predictable to observe different results in mice intestine administered with VSL#3 due to different cells involved in mice study in contrast to PBMC cell population.

In summary, increased expression of IL-8 on its own could singularly be indicative of inflammation, but in the context of all other upregulated anti-inflammatory cytokine and mediators found in this study, this may not be interpreted as an inflammatory effect. IL-8 upregulation might also be interpreted as requirement for the initial stimulatory effect of *S. thermophilus* 285 to switch on the immune responses by initiating innate immunity, which by the progress of immune response, expression of CCR5 (which in turn influences Th1 immune responses), as well as CXCL10 (induced by IFNγ) are reduced by *S. thermophilus* 285. This might be suggestive of modulation of immune responses by *S. thermophilus* 285 to keep the adaptive immune responses in check.

## 4.4. *S. thermophilus* 285 significantly upregulates mRNA expression level of colony stimulating factor

CSF (GM-CSF) is secreted by machrophages, NK cells and T cells, enables cell proliferation and differentiation, stimulates the production of various immune cells, in particular it

increases the production of machrophages which are important in fighting againts infections. CSF-2, is vastly increased (130 fold) by PBMC co-cultured with *S. thermophilus* 285 which is in alignment to our previous data whereby *S. thermophilus* 1275, *S. thermophilus* 1342 and *S. thermophilus* 285 induced U937 monocyte cell line to secrete high levels of GM-CSF with ST285 being the highest inducer [1]. Likewise, another study used RT$^2$ Profiler PCR Arrays for mouse cytokines and chemokines to demonstrate that *L. rhamnosus* GR-1 (GR-1) induced high levels of granulocyte CSF (G-CSF) mRNA (60-fold) to bone marrow-derived mouse macrophages [55]. Likewise, PBMC co-cultured with *B. infantis* 52486 significantly increases GM-CSF [56].

GM-CSF is generally accepted as an inflammatory cytokine, its inflammatory activity is primarily associated with its role as granulocytes and macrophages growth and differentiation factor. GM-CSF-mediated inflammation has also been associated with certain types of autoimmune diseases such as rheumatoid arthritis and multiple sclerosis. However, in many instances GM-CSF plays anti-inflammatory/regulatory roles; GM-CSF can modulate differentiation of DC to render them into tolerogenic DCs, which, can stimulate anti-inflammatory Treg cells [57]. In addition, either of pro-inflammatory or regulatory effects of GM-CSF appears to be dependent on the amount of CSF and the presence of other relevant cytokines in the context of an immune response. There is also evidence that G-CSF induces expansion of IL-10-producing cells [58]. Our results show very high overexpression of CSF, which might be suggestive of anti-inflammatory effect of *S. thermophilus* 285 on PBMC.

## 4.5. *S. thermophilus* 285 downregulates mRNA expression levels of toll-like receptors

Toll-like receptors (TLRs) recognize pathogen-associated molecular patterns (PAMPs) that are expressed on infectious bacteria and mediate the production of cytokines necessary for the development of effective immunity [59]. TLRs recognize pathogens and activate the innate immune responses. TLR-1, TLR-2, TLR-4 and TLR-8 are part of the innate immune response and are involved in defense against bacteria. Co-culturing *S. thermophilus* 285 with human PBMC downregulated the expression of TLR. Similarly, *E. coli* K88 and mycotoxin zearalenone (ZEA) infection of IPEC-1 epithelial cell line was protected in the presence of mixed Lactobacillus strains (*L. acidofilus* ID11692, *L. plantarum* ID1253 and *L. paracasei* ID13239) by downregulating TLR-1, TLR-2 and TLR-4 gene expression [60].

TLRs are critical in bacterial recognition and host defence, such as lipo-teichionic acid (LTA) and lipo-polysaccharide (LPS) from Gram-positive and Gram-negative bacteria respectively [61, 62]. Activation of some of these molecules and mediators like TLR (especially TLR-2 and TLR-4) arbitrates to pro-inflammatory actions and further defensive functions of innate immunity [63–65]. The TLR-2 and TLR-4 activation and expression by LPS (pathogens) is known as one of the most important mechanisms by which the immune system controls reactions to bacteria in particular in the activation phase, therefore, over-expression of TLR-2 and TLR-4 during any bacterial infection could cause an elevated inflammatory response in the body. While early activation of TLRs expression is reported in response to bacterial LPS from pathogenic *Salmonella typhimurium* [61] as well as *E. coli* infection in bovine intestinal epithelial cells [66], our results show tolerance as a result of co-culturing PBMC with *S. thermophilus* 285 by down regulation of TLRs genes.

Downregulated mRNA expression of TLRs genes, specifically TLR-1, TLR-2, TLR-4 and TLR-8 indicates anti-inflammatory characteristics for *S. thermophilus* 285. Given that TLR-1, TLR-2, TLR-4 and TLR-8 are members of the innate immune response and play key roles in the defense against bacteria, downregulation of TLRs could be suggestive of a protective

mechanism to keep *S. thermophilus* 285 safe by tolerance towards *S. thermophilus* 285. Perhaps designing experiments that allow different incubation period, as well as adding pathogenic bacteria to the co-cultured *S. thermophilus* 285-PBMC can help to illustrate if lesser coculture time and/or presence of pathogens can result in a shift towards upregulation of TLRs instead.

### 4.6. *S. thermophilus* 285 downregulates cell surface markers CD14, CD40, CD86

CD14, CD40 CD86 are expressed on the cell surface of monocytes, macrophages and DC. CD14 is expressed on the surface of monocytes and primarily binds to bacterial constituents [67–69]. We previously showed that U937 monocyte cell line exposed to *S. thermophilus* 1342, *S. thermophilus* 1275 or *S. thermophilus* 285 enhanced expression of CD14 after 24 and 48 hrs, and *S. thermophilus* 285 was the most potent at 48 hrs [1]. However, in bulk PBMC cultures, CD14 expression was significantly downregulated in the presence of *S. thermophilus* 285, which is in accordance with downregulation of TRLs in particular TRL-4. In other studies, the combination of 3 probiotics (*L. acidophilus*, *L. delbrueckii* ssp. *bulgaricus* and *B. bifidum*) stimulated increased expression of cell surface markers, CD14, CD80 and MHC class II [1]. *E. coli* Nissle 1917, widely used as a probiotic for the treatment of inflammatory bowel disorders, expresses a K5 capsule important in *E. coli* mediating interactions with intestinal epithelial cells and chemokine expression. *E. coli* Nissle 1917 has been shown to induce mRNA expression of CD14 by intestinal Caco-2 cells [70].

CD40 is a costimulatory protein on antigen presenting cells and is essential for their activation. CD40 is a key mediator in a wide range of inflammatory and immune responses and its gene expression was downregulated by PBMC in the presence of *S. thermophilus* 285. In previous experiments with U937 monocyte cell line, co-culture with *S. thermophilus* 1342, *S. thermophilus* 1275 or *S. thermophilus* 285, resulted in small increase in CD40 [1].

CD86 (B7-2) is expressed on APCs and delivers co-stimulatory signals required for the activation and survival of T cells. CD86 plays the role of the ligand for T cells external CD28, and CTLA-4 (CD28) in regulation and cell to cell dis-association. CD86 acts in conjunction with CD80 to prime Th cells, delivering opposing functions on Treg cells through CTLA-4 and T cell surface CD28 protein. Expression of CD86 by PBMC is downregulated significantly, suggesting an anti-inflammatory profile following exposure to *S. thermophilus* 285. *S. thermophilus* bacteria promote CD86 expression required for T cell activation and the maintenance of immune responses, CD86 downregulation by *S. thermophilus* 285 suggests a regulating and damping effect of *S. thermophilus* 285 on PBMC, being interpreted as immunomodulation of adaptive immunity [71]. We previously noted using U937 monocyte cell line in the presence of *S. thermophilus* 1342, *S. thermophilus* 1275 and *S. thermophilus* ST285 increased expression of CD86 [1]. Similarly, *L. plantarum* WCFS1 and *L. fermentum* GR1485 upregulate CD86 on monocytes, conversely, *L. rhamnosus* and *L. delbruekii* reduced its expression [72].

Additionally, monocytes isolated from PBMC and differentiated into immature DCs by GM-CSF and IL-4, and co-cultured with *B. breve* Bb99, *L. lactis* subsp. cremoris ARH74 and *S. thermophilus* THS also increase CD86 expression [23]. Another study used bone marrow-derived DCs from DQ8-transgenic mice and co-culture with *L. plantarum* and *L. paracasei* and *B. lactis* increases CD86 differentially with the highest CD86 being noted in co-administration of *L. plantarum* and *L. paracasei* [73]. The contrast between these studies to the findings herein could be due to the differences in the nature of studies; we co-cultured PBMC with *S. thermophilus* 285 bacteria only and the other studies used mouse bone marrow-derived DCs co-cultured with three different probiotics leading to predictable differences.

Given the downregulation of cell surface markers and their roles in immunity, CD14 (involved in innate immunity), CD40 (involved in innate immunity), and CD86 (T cell activation), following *S. thermophilus* 285 co-culture is suggestive of an anti-inflammatory anti-activation profile for *S. thermophilus* 285. In addition, as all these cell surface markers are interlinked with defence against bacteria either through innate or adaptive immune responses, downregulation of these markers could be suggestive of *S. thermophilus* 285 initiating self-tolerance via regulating immune responses, which in turn modulates the immune responses too.

## 4.7. *S. thermophilus* 285 differentially downregulates mRNA expression level of other innate and adaptive immune response markers and chemokines

Complement component 3 (C3) is associated with complement cascades in immune responses by enhancing antibody function, phagocytosis and stimulation of inflammation [74–76]. GATA3 transcriptome is also important in both humoral immunity and inflammatory responses. Downregulation of C3 gene expression and significant reduced expression of GATA3 transcriptome by PBMC co-cultured with *S. thermophilus* 285 in noted. Similarly, lipoteichoic acid (p-LTA) extracted from *L. plantarum* K8 inhibits C3 mRNA *in vitro* and *in vivo*. In human clinical studies, blocking GATA3 is able to control allergy responses, inflammatory diseases and asthma [77]. C3 and GATA3 downregulation suggests that *S. thermophilus* 285 is able to lower inflammation (C3), as well as being a viable candidate for further pre-clinical and clinical studies for the management of such diseases.

Interferon regulatory factor (IRF) 7, integrin alpha M (ITGAM), Lysozyme (LYZ) and NALP3 are other innate immune response factors. IRF7, a member of IRF family and present on monocytes, macrophages, granulocytes, and NK cells, and expressed predominantly in macrophages (a component of the inflammasome) [78]. IRF7 plays a role in the transcriptional activation of virus-inducible cellular genes, including the type I interferon genes. ITGAM is involved in a number of inflammatory responses (i.e. cell-mediated cytotoxicity, phagocytosis, and chemotaxis). LYZ acts as an antimicrobial enzyme present in neutrophils and macrophages. IRF7 gene regulation decreased considerably along with ITGAM gene expression, which is downregulated when PBMC are co-cultured with *S. thermophilus* 285. NALP3 and LYZ are downregulated markedly in co-culture of PBMC with *S. thermophilus* 285. However, in a previous study, we showed significant upregulation of CD11b (ITGAM) by monocytic U937 cells when co-cultured with *S. thermophilus* 1342, *S. thermophilus* 1275 and *S. thermophilus* 285 bacteria [1]. *S. thermophilus* 285-induced downregulation of IRF7, ITGAM, NALP3 and LYZ in PBMC, suggestive of an anti-inflammatory effect of *S. thermophilus* 285 on PBMC as well.

Non-receptor tyrosine-protein kinase (TYK2) is an enzyme [7] that contributes to adaptive immune responses due to its implication in IFNα, IL-6, IL-10 and IL-12 signaling, also involved in transducing signals of IL-6, IL-10 and IL-23. TYK2 gene expression is significantly downregulated in PBMC co-cultured with *S. thermophilus* 285, supporting an anti-inflammatory profile for *S. thermophilus* 285. In addition, myeloperoxidase (MPO), an enzyme abundantly expressed by neutrophils and promotes inflammation, is also involved in autoimmune disorders (multiple sclerosis, rheumatoid arthritis) [79, 80]. A decreased expression of MPO has been suggested to manage these autoimmune disorders by decreasing the inflammatory state. *S. thermophilus* 285 co-cultured with PBMC decreased the expression of MPO, suggestive of an anti-inflammatory benefit of *S. thermophilus* 285.

IFNAR1 is a type I membrane protein which is a receptor for IFNα and IFNβ involved in defence against viruses. IFNAR1 signaling is associated with pro-inflammatory cytokine

production [81]. In fact, IFNAR1 knockout mice show decreased pro-inflammatory cytokiens and chemokines [81]. IFNAR1 is significantly downregulated by PBMC following co-culture with ST285, supporting an anti-inflammatory role of *S. thermophilus* 285. In addition, SLC11A1 involved in T cell activation, is involved in inflammatory disorders such as autoimmune type 1 diabetes [82, 83], is downregulated by PBMC in the presence of *S. thermophilus* 285. Furthermore, the Beta-actin (ACTB) which stimulates eNOS and increase nitric oxide (NO) [84] involved in immunity and inflammation [85], is downregulated by PBMC following co-culture with *S. thermophilus* 285.

We determined the immune modulatory effects of *S. thermophilus* 285 to human PBMC and show that it has an array of anti-inflammatory immune-modulatory properties. *S. thermophilus* 285 decreases mRNA expression IL-18, IFNγR1, CCR5, CXCL10, TLR-1, TLR-2, TLR-4, TLR-8, CD14, CD40, CD86, C3, GATA3, ITGAM, IRF7, NLP3, LYZ, TYK2, IFNR1, and upregulates IL-1α, IL-1β, IL-6, IL-8, IL-10, IL-23, IFNγ, TNFα, CSF-2. No changes to mRNA expression are noted with IFNA1, IFNB1, IL-4, IL-5, IL-13, CCL2, CCL5, CCL8, CCR4, CCR8, CXCR3, TLR-3, TLR-5, TLR-6, TLR-9, CD4, CD80, FOXP3, STAT3, CD40LG, HLA-A, HLA-E, RORC. The data demonstrates a predominant anti-inflammatory profile exhibited by *S. thermophilus* 285, and further work is required to determine its effects in inflammatory disease models *in vitro* and *in vivo*, such as multiple sclerosis, inflammatory bowel disease and allergies. Future investigations using RNA-Seq and Western blots are some of the next logical steps to confirm and further investigate these results.

## 5. Conclusion

Probiotics are beneficial microorganism with immunomodulatory properties, which aid the maintenance of a healthy immune system. *S. thermophilus* is often used in fermented dairy products such as cheeses and yogurts and is believed to potentially have health benefits. We determined the immune modulatory effects of *S. thermophilus* 285 to human peripheral blood mononuclear cells and show that it has an array of anti-inflammatory immune-modulatory properties. *S. thermophilus* 285 decreases mRNA expression IL-18, IFN receptor, CCR5, CXCL10, TLR-1, TLR-2, TLR-4, TLR-8, CD14, CD40, CD86, C3, GATA3, ITGAM, IRF7, NLP3, LYZ, TYK2, IFNR1, and upregulates IL-1α, IL-1β, IL-6, IL-8, IL-10, IL-23, IFN-γ, TNF-α, CSF-2. No changes to mRNA expression were noted with IFNA1, IFNB1, IL-4, IL-5, IL-13, CCL2, CCL5, CCL8, CCR4, CCR8, CXCR3, TLR-3, TLR-5, TLR-6, TLR-9, CD4, CD80, FOXP3, STAT3, CD40LG, HLA-A, HLA-E, RORC.

## Supporting information

**S1 Table. List of the innate or adaptive genes and housekeeping genes; their symbols, full name and description that are investigated in in this study.**
(DOCX)

**S1 Fig. Presentation of data as a heatmap of average gene/RNA expressions of PBMC (n = 3) co-cultured with *S. thermophilus* ST285, compared to control.** Green represents down regulated genes to red represents upregulated genes.
(DOCX)

## Acknowledgments

All authors were supported by VU Research and in particular, Institute for Health and Sport and the Institute for Sustainable Industries and Liveable Cities at Victoria University.

## Author Contributions

**Conceptualization:** Narges Dargahi, Vasso Apostolopoulos.

**Data curation:** Narges Dargahi.

**Formal analysis:** Narges Dargahi.

**Investigation:** Narges Dargahi.

**Methodology:** Narges Dargahi, Joshua Johnson.

**Project administration:** Narges Dargahi, Vasso Apostolopoulos.

**Software:** Narges Dargahi, Joshua Johnson.

**Supervision:** Joshua Johnson, Vasso Apostolopoulos.

**Validation:** Narges Dargahi, Joshua Johnson, Vasso Apostolopoulos.

**Visualization:** Narges Dargahi.

**Writing – original draft:** Narges Dargahi.

**Writing – review & editing:** Narges Dargahi, Joshua Johnson, Vasso Apostolopoulos.

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
