## [Decision Letter · Decision Letter 0]

10 Oct 2019

PONE-D-19-20855

Streptococcus thermophilus alters the expression of genes associated with innate and adaptive immunity in human peripheral blood mononuclear cells

PLOS ONE

Dear Dr Dargahi,

Thank you for submitting your manuscript to PLOS ONE. After careful consideration, we feel that it has merit but does not fully meet PLOS ONE’s publication criteria as it currently stands. Therefore, we invite you to submit a revised version of the manuscript that addresses the points raised during the review process.

We would appreciate receiving your revised manuscript by Nov 24 2019 11:59PM. To enhance the reproducibility of your results, we recommend that if applicable you deposit your laboratory protocols in protocols.io, where a protocol can be assigned its own identifier (DOI) such that it can be cited independently in the future. For instructions see: http://journals.plos.org/plosone/s/submission-guidelines#loc-laboratory-protocols

We look forward to receiving your revised manuscript.

Kind regards,

Hossam M Ashour

Academic Editor

PLOS ONE

Journal Requirements:

2. Please amend either the abstract on the online submission form (via Edit Submission) or the abstract in the manuscript so that they are identical.

Additional Editor Comments (if provided):

Please address all comments raised by reviewers. There was also an extensive amount of self citation, which needs justification.

Reviewers' comments:

Reviewer's Responses to Questions

**Comments to the Author**

1. Is the manuscript technically sound, and do the data support the conclusions?

Reviewer #1: Partly

Reviewer #2: Yes

Reviewer #3: Yes

Reviewer #4: Partly

2. Has the statistical analysis been performed appropriately and rigorously? 

Reviewer #1: No

Reviewer #2: Yes

Reviewer #3: Yes

Reviewer #4: Yes

3. Have the authors made all data underlying the findings in their manuscript fully available?

Reviewer #1: Yes

Reviewer #2: Yes

Reviewer #3: Yes

Reviewer #4: Yes

4. Is the manuscript presented in an intelligible fashion and written in standard English?

Reviewer #1: Yes

Reviewer #2: Yes

Reviewer #3: Yes

Reviewer #4: Yes

5. Review Comments to the Author

Reviewer #1: The manuscripts presented by Dargahi et al presented data of only mRNA expression in the PMBC of donors to conclude the immune regulation by Streptococcus thermphilus. While the premise to study Streptococcus thermphilus is interesting because of their applications as probiotics, there are several major flaws in the experimental model used that must be addressed.

Major comments:

1. The relevance of the experimental model used needs to be justify. The authors used PBMC co-culture with ST285 as the primary system to explain the immune regulation properties of the bacteria. Streptococcus thermphilus is a probiotic, and they generally exist within the mucous layer of the gut and do not cross over to meet immune cells in the blood where the PBMC are harvested from. Importantly, there are a wide array of studies demonstrating that the immune cells in the gut differ from the PBMC both in terms of activation status and proportion of different subsets. The authors need to justify the use of PBMCs and limit conclusion to immune subsets which are comparable between the gut and the blood.

2. In this co-culture system, the control used is without the bacteria. It would be important to include a unrelated bacteria strain such as E.coli to emphasize the specificity of the response induced by ST285.

3. The entire paper relied solely on gross gene expression data from total PBMC. This lacks the specificity to understand the impact on specific cell type in the fraction because most cytokines could be produced by more than 1 cell type. In addition, there are no protein expression data to back up the conclusion. I would propose doing intracellular staining of cytokines in specific subsets through flowcytometry in the PBMC on the core cytokines identified from the gene expression study.

4. The presentation of the data needs to be improved. Particularly, the heat map in fig 1 is difficult to read. Usually, heat map data are presented using hierarchical clustering, with the genes on 1 axis and donors on another axis, to clearly display the differences between stimulated and unstimulated. Next, the author did one-way ANOVA analysis for all the figures, I assume its between different genes displayed in each panel. This is unnecessary since which genes goes into which panel is arbitrary. Instead, since each donor have a un-stimulated controls, the author should use wilcoxon-matched pairs signed rank test for each respective gene. Lastly, while the mean value of the unstimulated controls will be 1, they should still be plotted in to display the variation between them.

Reviewer #2: You had a conclusion as a ''upregulation of IL-1 alpha, IL-6, IL 10, and downregulation of IL-2, IL-18 and an absence of change in IL-17A (despite increase in IL-23 alpha) designates ST285 to possess anti inflammatory effects on human PBMC''.

Interleukin 6 (IL-6) is an interleukin that acts as both a pro-inflammatory cytokine and an anti-inflammatory myokine.Interleukin (IL)-6 is produced at the site of inflammation and plays a key role in the acute phase response as defined by a variety of clinical and biological features such as the production of acute phase proteins.

In general, Interleukin 1 alpha is responsible for the production of inflammation, as well as the promotion of fever and sepsis. We can say both of IL-1 and IL-6 have inflamatuarty affect. So that, It should be explained how you interpret the increase of IL-1 alpha and IL-6 syntheses as anti-inflammatory effect.

When writing bacterial names should be paid attention to Nomenclature of Microorganisms. Binary names, consisting of a generic name and a specific epithet (e.g., Escherichia coli), must be used for all microorganisms. Names of genus level may be used alone, but specific and subspecific epithets may not. A specific epithet must be preceded by a generic name, written out in full the first time it is used in a paper. Thereafter, the generic name should be abbreviated to the initial capital letter (e.g., E. coli), provided there can be no confusion with other genera used in the paper.

Reviewer #3: 1- Authors gave information about PBMC in the section of Materials and Methods which can be presented in the introduction section.

2- Authors did not write the number of the collected whole blood samples used in this study in the section of Materials and methods but they wrote it in the result section. (please write it in the materials and Methods section).

3- It is preferable to list the genes and housekeeping genes tested in this study (as supplementary file), their reaction conditions and their role as innate or adaptive to be helpful for any researcher in his study.

Reviewer #4: Dargahi et al. discusses the possible effects of Streptococcus thermophilus on innate and adaptive immune response using in vitro (ex vivo) PBMCs. The objective of this study according to the authors was to understand Streptococcus thermophilus anti-inflammatory and modulatory properties. Streptococcus thermophilus is used as probiotic and affects gut microbiome.

The manuscript is clear and well-written with detailed materials and methods section. However, I believe the authors could have consolidated the figures into 3 or 4 figures as the results section was lengthy describing just a single experiment.

There are drawbacks of this study, first of all, the whole study is based on a single experiment of isolation of PBMCs from buffy coats from blood bank and performing an expression mini-array (qPCR profiling array of 84 genes). To start with, if that's the only experiment done, authors should have tried to get better quality RNA by storing the lysed PBMCs (treated and untreated) in the lysis buffer of the provided manufacturer. If that was done, RNA quality would have been much better than going to RIN values of 7.5, they could have easily reached 10 with minimum of 8.5 (high quality RNA). Another thing is that other experiments could have been done to support the expression analysis data and take it a step further.

The authors drew too many conclusions from just qPCR profiling of the treated and untreated PBMCs. Additional supporting experiments should have been done.

6. PLOS authors have the option to publish the peer review history of their article (what does this mean?). If published, this will include your full peer review and any attached files.

Reviewer #1: Yes: Teck-Hui Teo

Reviewer #2: No

Reviewer #3: Yes: Rehab Mahmoud Abd El-Baky

Reviewer #4: No

---

## [Author Response · Author response to Decision Letter 0]

24 Nov 2019

Reviewers' comments:

Reviewer's Responses to Questions

Reviewer #1: The manuscripts presented by Dargahi et al presented data of only mRNA expression in the PMBC of donors to conclude the immune regulation by Streptococcus thermphilus. While the premise to study Streptococcus thermphilus is interesting because of their applications as probiotics, there are several major flaws in the experimental model used that must be addressed.

Major comments:

1. The relevance of the experimental model used needs to be justify. The authors used PBMC co-culture with ST285 as the primary system to explain the immune regulation properties of the bacteria. Streptococcus thermphilus is a probiotic, and they generally exist within the mucous layer of the gut and do not cross over to meet immune cells in the blood where the PBMC are harvested from. Importantly, there are a wide array of studies demonstrating that the immune cells in the gut differ from the PBMC both in terms of activation status and proportion of different subsets. The authors need to justify the use of PBMCs and limit conclusion to immune subsets which are comparable between the gut and the blood.

Answer: The Reviewer # 1 correctly refers to the natural habitat of the probiotics/ microflora inside human body, and the difference between circulating immune cells and the one in the gut. However, PBMC are some of the very common models which are commonly used to examine human immune response to different stimuli including those from bacteria and probiotics. Numerous cell types are involved in maintenance of the intestinal tissue. However, the main players are cells of the epithelial lining and the immune system. Human PBMCs are used to investigate the effect of probiotics, food bioactives and many different stimuli on various immune cells because these cells are easily isolated from blood of healthy donors or buffy coats (buffy coats are leukocyte concentrates; a by-product from anti-coagulated whole blood by Blood Banks in the manufacturing of red blood cell and thrombocyte concentrates). Even though PBMCs have a different composition, phenotype and activation status than cells found in intestinal tissue, utilising PBMC has been known as a valuable test system for investigation of immune modulatory effects of probiotic bacteria and many other compounds. For this reason using PBMC has been used in the current study as well as in many studies to determine immune-modulatory responses. In addition, there is an interaction between PBMCs and intestinal cells, hence PBMC can influence intestinal cells. In 2013 a paper published in PLOS One showed that there is interaction of PBMC with intestinal epithelial cells and that the probiotic Lactobacillus casei was able to modulate cytokine responses to human PBMC making them immunosuppressive (Tittanen, M et al., 2013) similar to the findings with ST285 in our studies. However, we have assessed an entire spectrum of cytokines and genes involved in the innate and adaptive immune response and determined the immune modulatory effects of ST285 to PBMC.

2. In this co-culture system, the control used is without the bacteria. It would be important to include a unrelated bacteria strain such as E.coli to emphasize the specificity of the response induced by ST285.

Answer: As reviewer accurately stated many studies use E.coli as positive control. E.coli has been proved over and over to trigger pro-inflammatory responses. In mRNA and gene expression studies, the tested/treated samples are always compared against a control which could be either positive control/E.coli or an untreated control samples. Comparing expression of genes against E.coli would provide a much larger up or down regulation of the changed genes, which could have led to an exaggerated up or down regulation of altered genes. Probiotic treated PBMC samples were compared against untreated ones which are the most common practice in PCR and specifically gene arrays (REF). In addition, in our previous publication, (Narges et al, Journal of Functional Foods 2018) we showed that LPS (an extract from E. coli) induced a pro-inflammatory profile to monocyte cell line. Here we demonstrate the effects of the probiotic bacteria ST285 effects on PBMC which induces a predominant anti-inflammatory profile. 

3. The entire paper relied solely on gross gene expression data from total PBMC. This lacks the specificity to understand the impact on specific cell type in the fraction because most cytokines could be produced by more than 1 cell type. In addition, there are no protein expression data to back up the conclusion. I would propose doing intracellular staining of cytokines in specific subsets through flowcytometry in the PBMC on the core cytokines identified from the gene expression study.

Answer: The reviewer raises a valid point. In fact, we previously determined 12 individual cell surface markers through flowcytometry and also studied secretion of 9 major cytokine and chemokines via bioplex (Dargahi et al, 2018). In these studies, the same probiotic (ST285) was used as in the current study. However, the aim of the current study was to determine a wide range of genes that covers most of cell surface markers, cytokine and chemokines to provide a bigger picture of immune modulatory properties of ST285. We have added a comment in the discussion to make our previous findings clearer.

4. The presentation of the data needs to be improved. Particularly, the heat map in fig 1 is difficult to read. Usually, heat map data are presented using hierarchical clustering, with the genes on 1 axis and donors on another axis, to clearly display the differences between stimulated and unstimulated. Next, the author did one-way ANOVA analysis for all the figures, I assume its between different genes displayed in each panel. This is unnecessary since which genes goes into which panel is arbitrary. Instead, since each donor have a un-stimulated controls, the author should use wilcoxon-matched pairs signed rank test for each respective gene. Lastly, while the mean value of the unstimulated controls will be 1, they should still be plotted in to display the variation between them.

Answer: The data presentation has now been improved for a better read of the heat map in fig S1. Also hierarchical clusters (fig1B) have now been added for a better and clearer presentation of obtained data.

The statistical analysis has been conducted using Qiagen web-based software (Qiagen RT2 profiler data analysis webportal) which is strongly recommended when using gene arrays. Then calculated the fold changes and analyzed data manually to compare results. Qiagen webportal uses internal controls that includes PCR array reproducibility control, RT efficiency control and genomic DNA contamination control, and ensures all arrays successfully passed all of these check point controls. Normalization of the raw data was performed using the included housekeeping genes (HKG) panel. In this system, means of 3 tested samples were compared against untreated.

The fold changes of unstimulated controls were calculated and considered as background expression, therefore those values have been deducted from the obtained gene expressions for tested samples to present real fold changes minus background (untreated controls). ). 

Wilcoxon is more appropriate for n ≥ 5 analysis and ANOVA is commonly used for this analysis. The confusion had arisen due to a lack of clarity in the methodology (it wasn’t conducted between genes, but between treatment vs control), which has been amended for clarity.

Reviewer #2: You had a conclusion as a ''upregulation of IL-1 alpha, IL-6, IL 10, and downregulation of IL-2, IL-18 and an absence of change in IL-17A (despite increase in IL-23 alpha) designates ST285 to possess anti inflammatory effects on human PBMC''.

Interleukin 6 (IL-6) is an interleukin that acts as both a pro-inflammatory cytokine and an anti-inflammatory myokine. Interleukin (IL)-6 is produced at the site of inflammation and plays a key role in the acute phase response as defined by a variety of clinical and biological features such as the production of acute phase proteins.

In general, Interleukin 1 alpha is responsible for the production of inflammation, as well as the promotion of fever and sepsis. We can say both of IL-1 and IL-6 have inflamatuarty affect. So that, It should be explained how you interpret the increase of IL-1 alpha and IL-6 syntheses as anti-inflammatory effect.

Answer: Reviewer #2 appropriately refers to the functions of IL-1a and IL-6. IL-1α (and IL-1β) has strong pro-inflammatory effect, however, it perform its roles via IL-1Ra (IL-1 receptor antagonist). The conclusion about the inti-inflammatory profile for ST285 is based on the overall effect of ST285 on PBMC. Despite the fact that the expression of IL-1a gene is increased, there is no increase in the expression for IL-1R which is required for the pro-inflammatory immune responses mediated by IL-1a and this may cause substantial reduction in reactions associated with IL-1a. Regarding IL-6, there has been a constant shifting in the pro- and anti-inflammatory dynamics and the equilibrium between them that leads to some controversy in research findings. IL-6 is one of the cytokines with both pro- and anti-inflammatory effects, whether IL-6 wears a pro- and anti-inflammatory mask all depends on the bio-environment. Once more, conclusion in this manuscript is driven based on the overall effect of ST285 bacteria and the results that support an anti-inflammatory environment caused by the bacterial stimulation. We have altered the discussion to include this reasoning for better clarity to the readers

When writing bacterial names should be paid attention to Nomenclature of Microorganisms. Binary names, consisting of a generic name and a specific epithet (e.g., Escherichia coli), must be used for all microorganisms. Names of genus level may be used alone, but specific and subspecific epithets may not. A specific epithet must be preceded by a generic name, written out in full the first time it is used in a paper. Thereafter, the generic name should be abbreviated to the initial capital letter (e.g., E. coli), provided there can be no confusion with other genera used in the paper. 

Answer: Although the binary nomenclature is the accurate and accepted way to state bacterial cultures, in some publications after the second time of writing abbreviated to the initial capital letter (e.g., E. coli), abbreviations are often used to save space and avoid repetition of names which are long. However, all the 2 latter abbreviations are replaced with the appropriate binary nomenclature as recommended by Reviewer #2.

Reviewer #3: 

1- Authors gave information about PBMC in the section of Materials and Methods which can be presented in the introduction section.

Answer: This information was removed from the material and methodology section and added to the introduction. 

2- Authors did not write the number of the collected whole blood samples used in this study in the section of Materials and methods but they wrote it in the result section. (please write it in the materials and Methods section).

Answer: The number of collected blood samples has now been added to the materials and Methods section

3- It is preferable to list the genes and housekeeping genes tested in this study (as supplementary file), their reaction conditions and their role as innate or adaptive to be helpful for any researcher in his study.

Answer: A list of the genes and housekeeping genes tested in this study has been added as a supplementary file, and their reaction conditions and their role in innate or adaptive immune responses has been briefly added as well (Table 1).

Reviewer #4: Dargahi et al. discusses the possible effects of Streptococcus thermophilus on innate and adaptive immune response using in vitro (ex vivo) PBMCs. The objective of this study according to the authors was to understand Streptococcus thermophilus anti-inflammatory and modulatory properties. Streptococcus thermophilus is used as probiotic and affects gut microbiome.

The manuscript is clear and well-written with detailed materials and methods section. However, I believe the authors could have consolidated the figures into 3 or 4 figures as the results section was lengthy describing just a single experiment.

Answer: the cytokine, chemokines and surface markers have been categorized and grouped based on their role in the immune system for a more cohesive description and better understanding of readers of non-immunological background. Due to reasons mentioned for systemic classification of the genes, combining 8 figures into 3-4 mixes up the results and discussion. Ideally it is best to keep the results and figures as they are. 

There are drawbacks of this study, first of all, the whole study is based on a single experiment of isolation of PBMCs from buffy coats from blood bank and performing an expression mini-array (qPCR profiling array of 84 genes). To start with, if that's the only experiment done, authors should have tried to get better quality RNA by storing the lysed PBMCs (treated and untreated) in the lysis buffer of the provided manufacturer. If that was done, RNA quality would have been much better than going to RIN values of 7.5, they could have easily reached 10 with minimum of 8.5 (high quality RNA). 

Answer: Human PBMC consist of sensitive cell populations, and, as blood samples are being processed for the purpose of PBMC isolation, they go through multiple centrifugation and washes and the process affects the cells. For this reason, and because of some level of ribonucleases produced by some of these cells, it is hard to obtain RNA with RIN 10. Although using cell lines and tissues obtaining RIN of 9.5-10 is much more feasible and we showed this in our previous publications. Any RNA of RIN 7 and above has been considered acceptable and enough high RNA quality. Several blood samples were used for PBMC isolation, treatment and extraction of RNA and only RNA samples with the highest RIN numbers (all above 8) were included for PCR. We have made this clearer in the material and methodology and results sections.

Another thing is that other experiments could have been done to support the expression analysis data and take it a step further.

The authors drew too many conclusions from just qPCR profiling of the treated and untreated PBMCs. Additional supporting experiments should have been done.

Answer: In our previous publications we showed immune modulatory effects using secreted cytokines using bioplex assays, and, flow cytometry of immune cell surface marker changes, in particular, monocyte cells. In the current study, we aimed to get a more comprehensive overview of the data, by undertaking an in depth gene array analysis of the effects of probiotics to human PBMC. We have altered the discussion to make this clearer to the readers. In this regard, we are currently determining the effects of probiotics using RNAseq and protein expression by western blot, and that data will be part of another publication. 

6. PLOS authors have the option to publish the peer review history of their article (what does this mean?). If published, this will include your full peer review and any attached files.

Answer: Yes we agree

Regards, 

Narges Dargahi 

Victoria University, Australia

---

## [Decision Letter · Decision Letter 1]

21 Jan 2020

Streptococcus thermophilus alters the expression of genes associated with innate and adaptive immunity in human peripheral blood mononuclear cells

PONE-D-19-20855R1

Dear Dr. Dargahi,

We are pleased to inform you that your manuscript has been judged scientifically suitable for publication and will be formally accepted for publication once it complies with all outstanding technical requirements.

With kind regards,

Hossam M Ashour

Academic Editor

PLOS ONE

Additional Editor Comments (optional):

Reviewers' comments:

Reviewer's Responses to Questions

**Comments to the Author**

1. If the authors have adequately addressed your comments raised in a previous round of review and you feel that this manuscript is now acceptable for publication, you may indicate that here to bypass the “Comments to the Author” section, enter your conflict of interest statement in the “Confidential to Editor” section, and submit your "Accept" recommendation.

Reviewer #2: All comments have been addressed

Reviewer #3: All comments have been addressed

Reviewer #4: All comments have been addressed

2. Is the manuscript technically sound, and do the data support the conclusions?

Reviewer #2: Yes

Reviewer #3: Yes

Reviewer #4: Partly

3. Has the statistical analysis been performed appropriately and rigorously? 

Reviewer #2: Yes

Reviewer #3: Yes

Reviewer #4: I Don't Know

4. Have the authors made all data underlying the findings in their manuscript fully available?

Reviewer #2: Yes

Reviewer #3: Yes

Reviewer #4: Yes

5. Is the manuscript presented in an intelligible fashion and written in standard English?

Reviewer #2: Yes

Reviewer #3: Yes

Reviewer #4: Yes

6. Review Comments to the Author

Reviewer #2: The authors have satisfactorily responded to all my questions and made the necessary changes to the manuscript.

Reviewer #3: (No Response)

Reviewer #4: Most of my comments have been addressed. However, I still think the supporting data is not enough for the conclusions, so it is up to the editor to see what fits publication in PLoS ONE.

7. PLOS authors have the option to publish the peer review history of their article (what does this mean?). If published, this will include your full peer review and any attached files.

Reviewer #2: Yes: Ali Acar

Reviewer #3: Yes: Rehab Mahmoud Abd El-Baky

Reviewer #4: No

---

## [Editor Report · Acceptance letter]

28 Jan 2020

PONE-D-19-20855R1 

*Streptococcus thermophilus* alters the expression of genes associated with innate and adaptive immunity in human peripheral blood mononuclear cells 

Dear Dr. Dargahi:

I am pleased to inform you that your manuscript has been deemed suitable for publication in PLOS ONE. Congratulations! Your manuscript is now with our production department. 

With kind regards,

on behalf of

Dr. Hossam M Ashour 

Academic Editor

PLOS ONE